# Is lack of health insurance a predictor of worsening of heart failure among adult patients attending referral hospitals in Northwestern Tanzania?

**Bahati M. K. Wajanga** [1,2]*, **Christine Yaeree Kim** [3‡], **Robert N. Peck** [1,2,4,5‡], **John Bartlett** [6,7,8‡], **Deodatus Mabula** [1,2☯], **Adinan Juma** [8‡], **Charles Muiruri** [6,7,8☯]

**1** Department of Internal Medicine, Bugando Medical Centre, Mwanza, Tanzania, **2** Department of Internal Medicine, Catholic University of Health and Allied Sciences–Bugando, Mwanza, Tanzania, **3** Department of Population Health Sciences, Duke University, Durham, NC, United States of America, **4** Division of Hospital Medicine, Department of Internal Medicine, Weill Cornell Medical College, New York City, NY, United States of America, **5** Department of Internal Medicine, Center for Global Health, Weill Cornell Medical College, New York City, NY, United States of America, **6** Duke Global Health Institute, Durham, NC, United States of America, **7** School of Medicine, Duke University School of Medicine, Durham, NC, United States of America, **8** Kilimanjaro Christian Medical University College, Moshi, Tanzania

☯ These authors contributed equally to this work.
‡ CYK, RNP, JB and AJ also contributed equally to this work.
* wajangabmk@gmail.com

**Data Availability Statement:** "All the data used to reach the conclusion drawn in the paper are

## Abstract

### Introduction

Health insurance coverage is critical for persons living with chronic conditions such as heart failure. Lack of health insurance may affect the ability to access regular healthcare appointments, pay for medication refills which can result in frequent hospitalization that is associated with poor clinical outcomes. In scarce resource locations such as sub-Saharan Africa, where uptake of health insurance is still suboptimal, the effect of health insurance on chronic conditions such as heart failure is poorly understood. The objective of this study was to assess the association of health insurance on the severity of heart failure for patients attending outpatient clinics at tertiary hospitals in Mwanza, Tanzania.

### Methods

As part of a larger cohort study, patients with heart failure were recruited from Bugando Medical Center (BMC) and Sekou Toure Regional Hospital (STRH) in Mwanza City, Tanzania. Heart failure was based on Framingham criteria and the severity was determined by New York Heart Association (NYHA) classification. Descriptive analysis and multivariable logistic regression were used to describe the study participants and to assess the association between health insurance status and the severity of heart failure at baseline.

### Results

418 patients were enrolled, and majority were female (n = 264, 63%), small scale farmers (n = 278, 66.5%) and were from Mwanza City (n = 299, 71.5%). More than two-thirds of

available at: https://www.kaggle.com/
charlesmuiruri/wanjanga-et-al-heart-failure.

**Funding:** Fogarty international Center, Award
Number: D43TW010138 The funders had no role
in study design, data collection and analysis,
decision to publish, or preparation of the
manuscript.

**Competing interests:** The authors have declared
that no competing interests exist.

patients did not have health insurance (n = 295, 70.6%) and the majority were in the NYHA I
and II classification (n = 267, 64.7%). There was no association between health insurance
status and the severity of heart (aOR 0.97; 95% CI 0.84–1.60). Being male, small-scale
businessperson and those seen at STRH was associated with higher odds of being in
NYHA Class III/IV (aOR = 1.97; 95% CI: 1.21–3.17), (aOR = 2.61; 95% CI: 1.27–5.34) and
(aOR 1.91 95% CI: 1.17–3.13) respectively. Having secondary and college education was
associated with lower odds of being in Class III/IV (0.42; 95% CI: 0.18–0.98) and (aOR =
0.23 95% CI: 0.06–0.86) respectively.

## Conclusion

In this study, only a third of the patients had health insurance. Health insurance was not
associated with the severity of heart failure. Since heart failure is a chronic condition patients
who do not have health insurance may incur out of pocket expenses, future research should
focus on the effect of out-of-pocket expenditures on clinical outcomes.

## Introduction

Cardiovascular disease (CVD) is the leading cause of morbidity and mortality globally,
accounting for 17.9 million deaths in 2016—or 44% of all deaths from non-communicable dis-
eases (NCDs) globally [1]. The burden of CVD is growing at an alarming pace in low- and
middle-income countries, especially in Sub-Saharan Africa (SSA) [1]. Among CVDs, heart
failure in SSA is associated with high rates of rehospitalization, poor quality of life and high
mortality [2, 3]. For example, prospective cohort studies of heart failure at two large hospitals
in Tanzania reported 1-year mortality rates of more than 20% [4, 5]. The burden of undiag-
nosed heart failure in SSA continues to rise, especially in rural areas; delayed diagnosis leads to
late presentation and greater the severity of disease which in turn creates more economic bur-
den for patients [6–8]. In SSA, it is estimated that 3–7% of all admissions to hospitals are due
to heart failure, and governments spent around 1% of their total budget for management of
heart failure [9, 10].

In countries without universal health coverage, costs of care for heart failure may expose
patients and their families to financial hardship [11]. Huffman et al. demonstrated that the
lack of health insurance for patients with heart failure resulted in catastrophic spending as
high as 92% of individual income. In this study, catastrophic spending was high in all income
groups and was associated with a two-fold increase for those in rural areas [12]. Health insur-
ance coverage may improve access to higher quality care with reduced out-of-pocket costs
[13]. In Tanzania, only 30% of the total population of 50 million people have access to health
insurance [14]. The most commonly used health insurance plan schemes are run by the gov-
ernment including Community Health Fund (CHF) and National Health Insurance (NHIF).
These health insurance plans cover 20% of general population. Private insurance plans (e.g.,
Strategy, Jubilee, AAR and Resolution) cover 1% of general population [15].

Hertz *et al.* reported that lack of insurance is likely to drive poor health-seeking habits and
care delays for patients with CVD [16]. Since heart failure is a chronic condition requiring reg-
ular follow up, uninsured patients may struggle to cover out of pocket costs like regular doctor
visits and lifelong medication refills, resulting in poor adherence and poor outcomes [17, 18].
However, the effect of health insurance on heart failure outcomes has not been well studied in

Tanzania. Identifying individual and societal predictors of heart failure outcomes is an important first step in the development of interventions to reduce heart failure morbidity and mortality. Thus, the objective of this study was to assess the association of patient characteristics and health insurance coverage on the severity of heart failure for patients.

## Methods

Participants of this study were recruited from hypertensive and cardiac Clinics of Bugando Medical Center (BMC) and Sekou Toure Regional Hospital (STRH). BMC is one of the 5 zonal referral hospitals in Tanzania with a catchment area of 14 million people. It is located in Mwanza City, the second largest urban center in Tanzania with 950 in-patient beds and approximately 300,000 hospitalizations per year. BMC has a dedicated cardiac clinic that cares for approximately 1400 patients every month. Sekou Toure Hospital is a regional referral hospital of Mwanza region with 300 beds capacity. On average, 320 patients are seen every month at the medical outpatient clinic at STRH. Adults with 18 years of age or older who had heart failure based on Framingham criteria and attending outpatient clinics at BMC and STRH who were fluent in Kiswahili and capable of providing written informed consent were enrolled in the study. During enrollment, a standardized questionnaire was administered in Kiswahili to capture the demographic data, insurance status, functional class of heart failure based on New York Heart Association (NYHA) criteria.

### Data analysis

Data were analyzed using Stata statistical software version 15 (StataCorp, College Station, TX). Our primary objective was to determine factors that contributed to severity of heart failure. We used NYHA classification to ascertain the severity of heart failure. The New York Heart Association (NYHA) Classification provides a simple way of classifying the extent of heart failure. It classifies patients in one of four categories based on their limitations of performing physical activity; the limitations/symptoms are in regard to normal breathing and varying degrees in shortness of breath and or angina pain. When the classes move from one to four, this means that the patient has an advanced stage of heart failure, as summarized below.

- Class I—No symptoms and no limitation in ordinary physical activity, e.g. shortness of breath when walking, climbing stairs etc.

- Class II—Mild symptoms (mild shortness of breath and/or angina) and slight limitation during ordinary activity.

- Class III—Marked limitation in activity due to symptoms, even during less-than-ordinary activity, e.g. walking short distances (20–100 m). Comfortable only at rest.

- Class IV—Severe limitations [19] We computed a binary outcome variable of the NYHA by merging Class I with Class II and Class III with Class IV. This categorization was deemed appropriate since higher NYHA functional class are predictive of poor outcome in patients with chronic heart failure compare to lower ones. Health insurance status was determined prior to diagnosis of heart failure. We described the study population using descriptive statistics: frequencies and proportions were calculated for categorical variables. Controlling for participants characteristics, multivariable logistic regression was used to assess the association between severity of heart failure (NYHA classification) and health insurance status. We used complete case analysis for missing data [22]. This means we only used cases in the dataset for which there were no missing values. All associations were presented as adjusted odds

ratios (aORs) with 95% confidence intervals (CIs). Statistical significance was set at a critical level of P < .05.

### Ethical issues

Ethical approval was obtained from the research and ethics committee of BMC (IRB certificate No. CREC/122/2016). All study participants were informed about the study by a study coordinator fluent in Kiswahili and provided written informed consent before participation

## Results

418 patients were enrolled in the cohort study. The majority of patients were female (n = 264, 63%), small scale farmers (n = 278, 66.5%), came from Mwanza City (n = 299, 71.5%). More than two-thirds of patients did not have health insurance (n = 295, 70.6%) and the majority were in the NYHA I and II classification (n = 267, 64.7%) as presented in Table 1 below.

We found no association between health insurance status and severity of heart failure as classified by the NYHA (aOR 0.97; 95% CI 0.84–1.60). Being male compared to female was

**Table 1. Demographic characteristics of 418 heart failure patients at BMC and STRH outpatient clinic in Mwanza, Tanzania.**

| Characteristic | Number(Percentage) |
|---|---|
| Sex | |
| Female | 264 (63%) |
| Male | 154 (37%) |
| Age | |
| 18–39 years | 51 (12.2%) |
| 40–64 years | 165 (39.5%) |
| > = 65 years | 202 (48.3%) |
| Education level | |
| No formal education | 86 (20.6%) |
| Primary education | 225 (53.8%) |
| Secondary education | 81 (19.3%) |
| College | 26 (6.2%) |
| Occupation | |
| Small- scale farmer | 278 (66.5%) |
| Small-scale business | 57 (13.6%) |
| Employed | 53 (12.7%) |
| Retired | 30 (7.2%) |
| Residence | |
| Mwanza | 299 (71.5%) |
| Outside Mwanza | 119 (28.5%) |
| Outpatient Clinic Site | |
| BMC | 261 (62.4%) |
| STRH | 157 (37.6%) |
| Health insurance Status | |
| Yes | 123 (29.4%) |
| No | 295 (70.6%) |
| NYHA Class | |
| I & II | 267 (64.7%) |
| III & IV | 146 (35.3%) |

**Table 2. Association between NYHA classification and health insurance status for heart failure patients in outpatient clinics at BMC and STRH in Mwanza, Tanzania.**

| Characteristics | NYHA classification | | Bivariate | | Multivariable | | |
|---|---|---|---|---|---|---|---|
| | I&II | III&IV | OR | 95%CI | aOR | 95%CI | P-value |
| Health insurance | | | | | | | |
| No | 182(62.5) | 109(37.5) | 1.0 | | 1.0 | | |
| Yes | 85(69.7) | 37(30.3) | 0.73 | 0.46–1.14 | 1.25 | 0.72–2.17 | 0.17 |
| Sex | | | | | | | |
| Female | 178(67.9) | 84(32.1) | 1.0 | | 1.0 | | |
| Male | 89(58.9) | 62(41.1) | 1.48 | 0.97–2.23 | 1.97* | 1.21–3.19 | 0.0065 |
| Age | | | | | | | |
| 18–39 years | 37(72.5) | 14(27.5) | 1.0 | | 1.0 | | |
| 40–64 years | 105(64.4) | 58(35.6) | 1.46 | 0.79–3.08 | 1.64 | 0.74–3.19 | |
| > = 65 years | 125(62.8) | 74(37.2) | 1.56 | 0.96–3.42 | 1.57 | 0.66–3.74 | 0.43 |
| Education level | | | | | | | |
| No formal education | 48(56.5) | 37(43.5) | 1.0 | | 1.0 | | |
| Primary education | 21(80.8) | 5(19.2) | 0.77 | 0.47–1.29 | 0.59 | 0.33–1.05 | |
| Secondary education | 139(62.6) | 83(37.4) | 0.46 | 0.24–0.89 | 0.42* | 0.18–0.98 | |
| College | 59(73.7) | 21(26.25) | 0.31 | 0.11–0.89 | 0.23* | 0.06–0.86 | 0.03* |
| Occupation | | | | | | | |
| Small- scale farmer | 175(62.6) | 100(36.4) | 1.0 | | 1.0 | | |
| Small-scale business | 29(51.8) | 27(48.2) | 1.63 | 0.91–2.90 | 2.61* | 1.27–5.34 | |
| Employed | 39(75.0) | 13(25.0) | 0.58 | 0.29–1.14 | 0.89 | 0.36–2.21 | |
| Retired | 24(80.0) | 6(20.0) | 0.44 | 0.17–1.10 | 0.62 | 0.21–1.86 | 0.02* |
| Outpatient clinic site | | | | | | | |
| BMC | 180(70.0) | 77(30.0) | 1.0 | | 1.0 | | |
| STRH | 87(55.8) | 69(44.2) | 1.85 | 1.23–2.80 | 1.91* | 1.17–3.13 | 0.003** |
| Residence | | | | | | | |
| Mwanza | 187(63.2) | 109(36.8) | 1.0 | | 1.0 | | |
| Outside Mwanza | 80(68.4) | 37(31.6) | 0.79 | 0.50–1.25 | 0.98 | 0.59–1.63 | 0.32 |

*** p<0.01

** p<0.05

* p<0.1 AOR: Adjusted Odd Ratio.

associated with higher odds of being in NYHA Class III/IV (aOR = 1.97; 95% CI: 1.21–3.17). Having secondary and college education compared to no formal education was associated with lower odds of being in Class III/IV (0.42; 95% CI: 0.18–0.98) and (aOR = 0.23; 95% CI: 0.06–0.86) respectively. Being small-scale businessperson compared to small-scale farmer was associated with higher odds being in NYHA Class III/IV (aOR = 2.61; 95% CI: 1.27–5.34). Heart failure patients seen at STRH had higher odds of being in Class III/IV (aOR 1.91; 95% CI: 1.17–3.13) as presented in Table 2 below.

## Discussion

Health insurance coverage is critical in reducing barriers to access, cost and quality of care [13]. Health insurance coverage may be even more crucial for patients with heart failure since these patients require close follow up in order to reduce morbidity and mortality. In this study, we did not find a significant difference in the severity of heart failure for those who had health insurance and those who did not. Mansi et al. found that the presence of health insurance or

type of coverage was not a significant predictor of any clinical outcomes for heart failure [20]. Our results may reflect the underlying contribution of behavioral and biological factors that are associated with adverse health outcomes. Pincus et al. observed that poor clinical outcomes resulted were better predicted by social conditions than access to care that is necessitated by health insurance coverage. In this study, other factors found to be associated with severity of heart failure included sex, education, occupation, and hospital site. Poorer heart failure outcomes have been associated with sociodemographic factors including gender, income level, employment status, and educational attainment. Male gender compared to female was associated with higher odds of having severe form of heart failure. This is not surprising since men are less likely to utilize healthcare services than women [22, 23]. This denies them an opportunity for early control of common cardiovascular diseases like hypertension, resulting in later complications like advanced heart failure. Future research should focus on interventions targeted to men who have heart failure. Having secondary and college education was associated with lower odds of having more severe heart failure. This association between education and heart failure outcomes has been observed in other studies from outside sub Saharan Africa [24]. A higher level of education may assist patient understanding of the need for adherence to heart failure medication and adherence to care, and this may in turn slow the progression of disease. Future studies should focus on tailoring interventions to accommodate patients with low levels of education.

Finally, being seen at the referral hospital (BMC) compared to the community hospital had higher odds of having severe heart failure. This is because BMC has more organized cardiac clinic with modern equipment and enough expert to take care of patients with cardiovascular conditions compared to Sekou Toure regional referral hospital.

These findings should be interpreted in light of the study's limitations. First, although combining NYHA I with II and III with IV may have meaningful clinical application, these groupings may have obscured the unobserved characteristics between different classes and therefore may have affected the efficiency of the estimator. Second, study participants came from a single district in Tanzania and therefore results may not be generalizable to the entire country. Finally, our results may have been confounded due to the lack of time since initial diagnosis of heart failure. This is because heart failure is likely to progress in severity over time. Notwithstanding these limitations, our findings have policy implications. Our results suggest that a focus on health insurance coverage to increase access to healthcare may not lead to better outcomes for heart failure. Indeed, interventions to reduce heart failure morbidity may need to focus on other patient characteristics, such as health literacy and sex, which appear to be important determinants of health. Contextual factors of specific healthcare facilities may also determine heart failure outcomes. As policy makers consider universal health coverage for their citizens, consideration should be given to other factors that facilitate optimal health.

## Acknowledgments

We appreciate the support provided by the Administration of Bugando Referral, Consultant and University Teaching Hospital, and Sekou Toure Regional Referral Hospitals We are grateful to the research team, Dr. Alphonce Ngerecha and Dr. Charles Kitundu.

## Author Contributions

**Conceptualization:** Bahati M. K. Wajanga, Robert N. Peck, John Bartlett, Deodatus Mabula, Charles Muiruri.

**Data curation:** Bahati M. K. Wajanga.

**Formal analysis:** Bahati M. K. Wajanga, Christine Yaeree Kim, Adinan Juma.

**Methodology:** Bahati M. K. Wajanga, Deodatus Mabula, Charles Muiruri.

**Project administration:** Bahati M. K. Wajanga.

**Supervision:** Bahati M. K. Wajanga.

**Validation:** Bahati M. K. Wajanga.

**Writing – original draft:** Bahati M. K. Wajanga.

**Writing – review & editing:** Bahati M. K. Wajanga, Christine Yaeree Kim, Robert N. Peck, John Bartlett, Deodatus Mabula, Adinan Juma, Charles Muiruri.

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
