## [Decision Letter · Decision Letter 0]

5 Nov 2021

PONE-D-21-28263Is Lack Of Health Insurance Associated With The Severity Of Heart Failure In Northwestern Tanzania?PLOS ONE

Dear Dr. Wajanga,

Thank you for submitting your manuscript to PLOS ONE. After careful consideration, we feel that it has merit but does not fully meet PLOS ONE’s publication criteria as it currently stands. Therefore, we invite you to submit a revised version of the manuscript that addresses the points raised during the review process.

We look forward to receiving your revised manuscript.

Kind regards,

Sreeram V. Ramagopalan

Academic Editor

PLOS ONE

2. Please consider changing the title so as to meet our title format requirement (https://journals.plos.org/plosone/s/submission-guidelines). In particular, the title should be "Specific, descriptive, concise, and comprehensible to readers outside the field" and in this case it is not informative and specific about your study's scope, methodology, and findings.

4. Thank you for stating the following in the Acknowledgments Section and Funding Sections of your manuscript:

Acknowledgement:

“The authors wish to thank MEPIJF Tanzania for its financial support. We appreciate the support provided by the Administration of Bugando Referral, Consultant and University Teaching Hospital, and Sekou Toure Regional Referral Hospitals We are grateful to the research team, Dr. Alphonce Ngerecha and Dr. Charles Kitundu.”

Funding:

“Research reported in this publication was supported by the Fogarty International Center (and list ALL co-funding partners) of the National Institutes of Health under Award Number D43 TW010138.  The contentis solely the responsibility of the authors and does not necessarily represent the official views of the National Institutes of Health.”

“This study was funded by The Medical Education Partnership Initiative-Tanzania (MEPI-T) a collaboration between Weill Cornell Medical College, Duke University, the Kilimanjaro Christian Medical University (KCMU College), and the Weill-Bugando Medical College of the Catholic University of Health and Allied Sciences (CUHAS) to support research training of junior faculty.

NO- The funders had no role in study design, data collection and analysis, decision to publish, or preparation of the manuscript.”

5. PLOS requires an ORCID iD for the corresponding author in Editorial Manager on papers submitted after December 6th, 2016. Please ensure that you have an ORCID iD and that it is validated in Editorial Manager. To do this, go to ‘Update my Information’ (in the upper left-hand corner of the main menu), and click on the Fetch/Validate link next to the ORCID field. This will take you to the ORCID site and allow you to create a new iD or authenticate a pre-existing iD in Editorial Manager. Please see the following video for instructions on linking an ORCID iD to your Editorial Manager account: https://www.youtube.com/watch?v=_xcclfuvtxQ. 

7. In your Data Availability statement, you have not specified where the minimal data set underlying the results described in your manuscript can be found. PLOS defines a study's minimal data set as the underlying data used to reach the conclusions drawn in the manuscript and any additional data required to replicate the reported study findings in their entirety. All PLOS journals require that the minimal data set be made fully available. For more information about our data policy, please see http://journals.plos.org/plosone/s/data-availability.

Reviewers' comments:

Reviewer's Responses to Questions

**Comments to the Author**

1. Is the manuscript technically sound, and do the data support the conclusions?

Reviewer #1: Yes

2. Has the statistical analysis been performed appropriately and rigorously? 

Reviewer #1: Yes

3. Have the authors made all data underlying the findings in their manuscript fully available?

Reviewer #1: No

4. Is the manuscript presented in an intelligible fashion and written in standard English?

Reviewer #1: Yes

5. Review Comments to the Author

Reviewer #1: The study investigates the relationship between health insurance coverage and severity of heart failure based on a sample of 418 patients enrolled at two outpatient clinics in Mwanza, Tanzania. Using multivariable logistic regression, no association was found between having health insurance and heart failure severity (NYHA classification III-IV vs. I-II). However, heart failure severity was associated with male sex, lower educational level, small scale business occupation, and treating clinic. The authors discuss how the results may reflect underlying social and biological factors contributing to clinical outcomes, however, on the question of health insurance coverage, call for further research into the impact of out-of-pocket expenses to assess what impact this has on clinical outcomes.

Overall, the reporting of the article is clear, well written and the interpretation of findings are reasonable and generally supported by the data.

Some points for consideration –

1.The analysis is cross-sectional in design, and the relationship between the exposure (health insurance) and outcome (heart failure severity) is considered, without mention of the temporality between the two variables. It would help to clarify –

Whether patients with insurance, had insurance, before the diagnosis of heart failure, or whether its possible some patients could have taken out insurance after the diagnosis of heart failure?

To what extent the population comprises patients with prevalent pre-existing heart failure or patients with incident new diagnoses of heart failure? Since heart failure is likely to progress in severity over time, I would suspect time since diagnosis might be an important confounding factor of the outcome to consider.

2.In the Data Analysis section, I would suggest providing more details on the NYHA score to assist readers who are unfamiliar with the measure i.e. the range of the score, and how it is interpreted, i.e. class I = no symptoms/limitations up to class IV= severe symptoms.

3.There is no mention of missing data, the extent and how this was handled in analysis – presume complete case analysis was used. Furthermore in table 2, ideally two columns should be added for the number of patients and events per category to assess whether null associations could be due to insufficient endpoints/power.

4.Line 149-150: “Pincus, et al observed that poor clinical outcomes resulted were better predicted by social conditions than access to care that is necessitated by health insurance coverage.” Check sentence for typo

6. PLOS authors have the option to publish the peer review history of their article (what does this mean?). If published, this will include your full peer review and any attached files.

Reviewer #1: No

---

## [Author Response · Author response to Decision Letter 0]

21 Dec 2021

1. PLOS ONE STYLE OF MANUSCRIPT

Please ensure that your manuscript meets PLOS ONE's style requirements, including those for file naming. The PLOS ONE style templates

Response

Thank you for your feedback. We have revised the current version to meet the PLOS ONE’s style requirement.

2. CHANGING OF TITLE

Please consider changing the title so as to meet our title format requirement (https://journals.plos.org/plosone/s/submission-guidelines). In particular, the title should be "Specific, descriptive, concise, and comprehensible to readers outside the field" and in this case it is not informative and specific about your study's scope, methodology, and findings

Response: 

We appreciate your guidance and have revised the current title to read 

“Is lack of Heath Insurance a predictor of worsening of heart failure among adult patients attending referral hospitals in Northwestern Tanzania?”

The short title is “Association of health insurance as a predictor of worsening heart failure”

3. GRANT INFORMATION

We note that the grant information you provided in the ‘Funding Information’ and ‘Financial Disclosure’ sections do not match.

Response: 

We apologize for the mistake and have provided the correct grant number in the funding information

4. ACKNOWLEDGEMENT AND FUNDING SECTIONS

We note that you have provided additional information within the Acknowledgements Section that is not currently declared in your Funding Statement. Please note that funding information should not appear in the Acknowledgments section or other areas of your manuscript. We will only publish funding information present in the Funding Statement section of the online submission form

Please remove any funding-related text from the manuscript and let us know how you would like to update your Funding Statement.

Response: 

Thank you for feedback. We have removed the funding information in the manuscript as per your instruction.

5. ORCID ID

PLOS requires an ORCID iD for the corresponding author in Editorial Manager on papers submitted after December 6th, 2016. Please ensure that you have an ORCID iD and that it is validated in Editorial Manager

Response:

Thank you for the feedback. I have included my ORCID ID, which is 0000-0002-1922-0746

6. ETHICS STATEMENT

Your ethics statement should only appear in the Methods section of your manuscript. If your ethics statement is written in any section besides the Methods, please move it to the Methods section and delete it from any other section. Please ensure that your ethics statement is included in your manuscript, as the ethics statement entered into the online submission form will not be published alongside your manuscript

Response: 

In the current version of the manuscript, we have included the ethics statement in the methods section on line 129-131

7. DATA AVAILABILITY STATEMENT

In your Data Availability statement, you have not specified where the minimal data set underlying the results described in your manuscript can be found. PLOS defines a study's minimal data set as the underlying data used to reach the conclusions drawn in the manuscript and any additional data required to replicate the reported study findings in their entirety. All PLOS journals require that the minimal data set be made fully available

Upon re-submitting your revised manuscript, please upload your study’s minimal underlying data set as either Supporting Information files or to a stable, public repository and include the relevant URLs, DOIs, or accession numbers within your revised cover letter

Response

Thank you for your feedback. The data set have been uploaded and URL included in the cover letter

8. REFERENCE LIST

Please review your reference list to ensure that it is complete and correct. If you have cited papers that have been retracted, please include the rationale for doing so in the manuscript text, or remove these references and replace them with relevant current references. Any changes to the reference list should be mentioned in the rebuttal letter that accompanies your revised manuscript. If you need to cite a retracted article, indicate the article’s retracted status in the References list and also include a citation and full reference for the retraction notice

Response: 

We reviewed the reference list and have an updated one in this version.

Reviewer #1

The study investigates the relationship between health insurance coverage and severity of heart failure based on a sample of 418 patients enrolled at two outpatient clinics in Mwanza, Tanzania. Using multivariable logistic regression, no association was found between having health insurance and heart failure severity (NYHA classification III-IV vs. I-II). However, heart failure severity was associated with male sex, lower educational level, small scale business occupation, and treating clinic. The authors discuss how the results may reflect underlying social and biological factors contributing to clinical outcomes, however, on the question of health insurance coverage, call for further research into the impact of out-of-pocket expenses to assess what impact this has on clinical outcomes.

Overall, the reporting of the article is clear, well written and the interpretation of findings are reasonable and generally supported by the data.

Some points for consideration –

1.The analysis is cross-sectional in design, and the relationship between the exposure (health insurance) and outcome (heart failure severity) is considered, without mention of the temporality between the two variables. It would help to clarify –

Whether patients with insurance, had insurance, before the diagnosis of heart failure, or whether its possible some patients could have taken out insurance after the diagnosis of heart failure?

Response: 

Thank you for your feedback. For this study, we asked about insurance status at the beginning of the study. The patients in this study had insurances prior the diagnosis of heart failure. We have added the description on lines 121-122

To what extent the population comprises patients with prevalent pre-existing heart failure or patients with incident new diagnoses of heart failure? Since heart failure is likely to progress in severity over time, I would suspect time since diagnosis might be an important confounding factor of the outcome to consider. 

Response: 

We agree that time since diagnosis of heart failure might be an important confounding factor. Since we did not collect this information in this study, we have declared this limitation on page line 187-189 variable, but then this would work best in a known cohort, followed over time, which was not in our case where the subject was seen once.

2. In the Data Analysis section, I would suggest providing more details on the NYHA score to assist readers who are unfamiliar with the measure i.e. the range of the score, and how it is interpreted, i.e. class I = no symptoms/limitations up to class IV= severe symptoms.

Response: 

We have provided more details about NYHA in the data analysis section lines 106- 118

3. There is no mention of missing data, the extent and how this was handled in analysis – presume complete case analysis was used. Furthermore in table 2, ideally two columns should be added for the number of patients and events per category to assess whether null associations could be due to insufficient endpoints/power.

Response: 

We included a description of how we handled missing data on lines 125-127

We appreciate your feedback and added the two columns.

4. Line 149-150: “Pincus, et al observed that poor clinical outcomes resulted were better predicted by social conditions than access to care that is necessitated by health insurance coverage.” Check sentence for typo

Response

Thank you. We have corrected the typo.

6. PLOS authors have the option to publish the peer review history of their article (what does this mean?). If published, this will include your full peer review and any attached files. If you choose “no”, your identity will remain anonymous but your review may still be made public

Response

We do not need to publish peer review history.

---

## [Editor Report · Decision Letter 1]

9 Feb 2022

Is lack of Heath Insurance a predictor of worsening of heart failure among adult patients attending referral hospitals in Northwestern Tanzania?

PONE-D-21-28263R1

Dear Dr. Wajanga,

We’re pleased to inform you that your manuscript has been judged scientifically suitable for publication and will be formally accepted for publication once it meets all outstanding technical requirements.

Kind regards,

Sreeram V. Ramagopalan

Academic Editor

PLOS ONE
---

## [Editor Report · Acceptance letter]

18 Feb 2022

PONE-D-21-28263R1 

*Is lack of Heath Insurance a predictor of worsening of heart failure among adult patients attending referral hospitals in Northwestern Tanzania?*

Dear Dr. Wajanga:

I'm pleased to inform you that your manuscript has been deemed suitable for publication in PLOS ONE. Congratulations! Your manuscript is now with our production department. 

Kind regards, 

on behalf of

Dr. Sreeram V. Ramagopalan 

Academic Editor

PLOS ONE